# Conserved crosstalk between histone deacetylation and H3K79 methylation generates DOT1L-dose dependency in HDAC1-deficient thymic lymphoma

Hanneke Vlaming[1,†,‡] (ID), Chelsea M McLean[1,†], Tessy Korthout[1], Mir Farshid Alemdehy[2], Sjoerd Hendriks[1], Cesare Lancini[1], Sander Palit[1], Sjoerd Klarenbeek[3], Eliza Mari Kwesi-Maliepaard[1], Thom M Molenaar[1], Liesbeth Hoekman[3], Thierry T Schmidlin[4], AF Maarten Altelaar[4,5] (ID), Tibor van Welsem[1], Jan-Hermen Dannenberg[1,§], Heinz Jacobs[2*] (ID) & Fred van Leeuwen[1,**] (ID)

## Abstract

DOT1L methylates histone H3K79 and is aberrantly regulated in MLL-rearranged leukemia. Inhibitors have been developed to target DOT1L activity in leukemia, but cellular mechanisms that regulate DOT1L are still poorly understood. We have identified the histone deacetylase Rpd3 as a negative regulator of budding yeast Dot1. At its target genes, the transcriptional repressor Rpd3 restricts H3K79 methylation, explaining the absence of H3K79me3 at a subset of genes in the yeast genome. Similar to the crosstalk in yeast, inactivation of the murine Rpd3 homolog HDAC1 in thymocytes led to an increase in H3K79 methylation. Thymic lymphomas that arise upon genetic deletion of *Hdac1* retained the increased H3K79 methylation and were sensitive to reduced DOT1L dosage. Furthermore, cell lines derived from *Hdac1^{Δ/Δ}* thymic lymphomas were sensitive to a DOT1L inhibitor, which induced apoptosis. In summary, we identified an evolutionarily conserved crosstalk between HDAC1 and DOT1L with impact in murine thymic lymphoma development.

**Keywords** Chromatin; H3K79 methylation; histone acetylation; histone ubiquitination; lymphoma

**Subject Categories** Cancer; Chromatin, Epigenetics, Genomics & Functional Genomics

The EMBO Journal (2019) 38: e101564

## Introduction

Aberrant histone modification patterns have been observed in many diseases, and this deregulation of chromatin can play a causative role in disease. Since epigenetic alterations are, in principle, reversible in nature, histone (de)modifiers are attractive therapeutic targets (Brien *et al*, 2016; Jones *et al*, 2016; Shortt *et al*, 2017). Several epigenetic drugs are currently in the clinic or in clinical trials, but for many of the drug targets, we are only beginning to understand their cellular regulation.

The histone H3K79 methyltransferase DOT1L (KMT4; Dot1 in yeast) is an epigenetic enzyme for which inhibitors are in clinical development for the treatment of MLL-rearranged (MLL-r) leukemia (Stein & Tallman, 2016). In MLL-r leukemia, DOT1L recruitment to MLL target genes, such as the HoxA cluster, leads to aberrant H3K79 methylation and increased transcription (reviewed in Vlaming & Van Leeuwen, 2016). Although the DOT1L inhibitor Pinometostat (EPZ-5676) has shown promising results in the laboratory and is currently in clinical development (Bernt *et al*, 2011; Daigle *et al*, 2013; Waters *et al*, 2015; Stein & Tallman, 2016; Stein *et al*, 2018), the cellular mechanisms and consequences of DOT1L deregulation are only just being uncovered (Vlaming & Van Leeuwen, 2016).

An important mechanism of regulation is the trans-histone crosstalk between monoubiquitination of the C-terminus of histone H2B (H2Bub) at lysine 120 (123 in yeast) and methylation of histone H3K79 (Zhang *et al*, 2015). The addition of a ubiquitin peptide to the nucleosome at this position occurs in a co-transcriptional manner and promotes the activity of Dot1/DOT1L, possibly by activation of DOT1L or coaching it toward H3K79 and thereby increasing the

1 Division of Gene Regulation, Netherlands Cancer Institute, Amsterdam, The Netherlands
2 Division of Tumor Biology & Immunology, Netherlands Cancer Institute, Amsterdam, The Netherlands
3 Experimental Animal Pathology, Netherlands Cancer Institute, Amsterdam, The Netherlands
4 Biomolecular Mass Spectrometry and Proteomics, Bijvoet Center for Biomolecular Research, Utrecht Institute for Pharmaceutical Sciences, Utrecht University and Netherlands Proteomics Centre, Utrecht, The Netherlands
5 Proteomics Facility, Netherlands Cancer Institute, Amsterdam, The Netherlands
  *Corresponding author. Tel: +31 20 5122065; E-mail: h.jacobs@nki.nl
  **Corresponding author. Tel: +31 20 5121973; E-mail: fred.v.leeuwen@nki.nl
  †These authors contributed equally to this work.
  ‡Present address: Department of Biological Chemistry and Molecular Pharmacology, Harvard Medical School, Boston, MA, USA
  §Present address: Genmab B.V., Antibody Sciences, Utrecht, The Netherlands

chance of a productive encounter (Vlaming *et al*, 2014; Zhou *et al*, 2016). Another mechanism of regulation is mediated by the direct interactions of DOT1L with central transcription elongation proteins (reviewed in Vlaming & Van Leeuwen, 2016). These interactions target DOT1L to transcribed chromatin and provide an explanation for the aberrant recruitment of DOT1L by oncogenic MLL fusion proteins (Deshpande *et al*, 2014; Li *et al*, 2014; Chen *et al*, 2015; Kuntimaddi *et al*, 2015; Wood *et al*, 2018). Further characterizing the regulatory network of DOT1L could lead to the identification of alternative drug targets for diseases in which DOT1L is critical and provide alternative strategies in case of resistance to treatment with DOT1L inhibitors (Campbell *et al*, 2017).

In a previous study, we presented a ChIP-barcode-seq screen (Epi-ID) identifying novel regulators of H3K79 methylation in yeast (Vlaming *et al*, 2016). The Rpd3-large (Rpd3L) complex was identified as an enriched complex among the candidate negative regulators of H3K79 methylation of a barcoded reporter gene. Rpd3 is a class I histone deacetylase (HDAC) that removes acetyl groups of histones, as well as numerous non-histone proteins, and is generally associated with transcriptional repression (Yang & Seto, 2008). Several inhibitors of mammalian HDACs have been approved for the treatment of cutaneous T-cell lymphoma and other hematologic malignancies, while others are currently being tested in clinical trials (West & Johnstone, 2014). HDAC1 and HDAC2, prominent members of the class I HDACs, are found in the repressive Sin3, NuRD, and CoREST complexes (Yang & Seto, 2008). Loss or inhibition of HDAC1/Rpd3 leads to increased histone acetylation, which in turn can lead to increased expression of target genes and cryptic transcripts (Carrozza *et al*, 2005; Joshi & Struhl, 2005; Li *et al*, 2007; Rando & Winston, 2012; Brocks *et al*, 2017; McDaniel & Strahl, 2017).

Here, we demonstrate that Rpd3 restricts H3K79 methylation at its target genes. Most euchromatic genes in the yeast genome are marked by high levels of H3K79me3. We observed that a subset of the genes that do not follow this pattern has lower H3K79me3 levels due to the action of the Rpd3L complex, which deacetylates its targets and imposes strong transcriptional repression and absence of H2Bub1. Importantly, the Rpd3-Dot1 crosstalk is conserved in mammals: Genetic ablation of *Hdac1* in murine thymocytes also leads to an increase in H3K79 methylation *in vivo*. High H3K79me is maintained in the lymphomas these mice develop, and a reduction in DOT1L activity by heterozygous deletion of *Dot1L* reduces tumor burden, an effect that was not observed upon homozygous deletion of *Dot1L*. Furthermore, DOT1L inhibitors induce apoptosis in *Hdac1*-deficient but not *Hdac1*-proficient thymic lymphoma cell lines, suggesting a DOT1L-dose dependence. Taken together, our studies reveal a new, evolutionarily conserved mechanism of H3K79me regulation by Rpd3/HDAC1 with relevance for cancer development.

# Results

## Identification of the Rpd3L complex as a negative regulator of H3K79 methylation

We recently reported a systematic screening strategy called Epi-ID to identify regulators of H3K79 methylation (Vlaming *et al*, 2016). In that screen, relative H3K79 methylation (H3K79me) levels at two

DNA barcodes (UpTag and DownTag) flanking a reporter gene were measured in a genome-wide library of barcoded deletion mutants, thus testing thousands of genes for H3K79me regulator activity at these loci (Fig 1A). Since higher Dot1 activity in yeast leads to a shift from lower (me1) to higher (me3) methylation states (Frederiks *et al*, 2008), the H3K79me3 over H3K79me1 ratio was used as a measure for Dot1 activity. A growth-corrected H3K79me score was calculated to account for the effect of growth on H3K79 methylation, and groups of positive and negative candidate regulators were identified (Vlaming *et al*, 2016). Components of the Rpd3L complex were enriched among candidate negative regulators (10-fold over-representation, $P = 1.2E-4$; Vlaming *et al*, 2016). The histone deacetylase Rpd3 is found in two complexes, the large Rpd3L complex and the small Rpd3S complex, which also share the subunits Sin3 and Ume1 (Carrozza *et al*, 2005; Keogh *et al*, 2005). A closer inspection of the Rpd3 complexes revealed that deletion of Rpd3L subunits resulted in an increase in H3K79 methylation on both the UpTag and the DownTag (promoter and terminator context, respectively; Fig 1B), with the exception of two accessory subunits that play peripheral roles (Lenstra *et al*, 2011). Deletion of the two Rpd3S-specific subunits did not lead to an increase in H3K79me (Fig 1B), which is consistent with Rpd3S binding and acting on coding sequences (Drouin *et al*, 2010) and thus away from the intergenic barcodes. To validate the effect on a global scale, we performed targeted mass spectrometry analysis to determine the relative levels of the different H3K79me states (me0 to me3) in *rpd3Δ* and *sin3Δ* strains. On bulk histones, these strains showed an increase in H3K79me (increase in H3K79me3 at the cost of lower methylation states; Fig 1C). The H3K79me increase was not caused by an increase in Dot1 protein (Fig EV1A) or mRNA expression (Kemmeren *et al*, 2014). Thus, although these regulators were identified using only two 20-base-pair barcodes to read out H3K79me levels at a reporter locus, their effects could be validated globally.

## Rpd3 represses H3K79 methylation at the 5′ ends of a subset of genes

We next asked at which regions Rpd3 and Sin3 regulate H3K79 methylation in yeast, other than the barcoded reporter gene. To address this, we performed ChIP-seq analysis for H3K79me1, H3K79me3, and H3 in wild-type and *rpd3Δ* strains. In addition, we included ChIP-seq for H2B and H2Bub using a site-specific antibody that we recently developed (Van Welsem *et al*, 2018). First, we considered the patterns in the wild-type strain. Both the coverage at one representative locus and across all genes in a heatmap showed that H3K79me3 is predominantly present throughout coding sequences of most genes, where H2Bub is also high, as reported previously (Figs 1D and E, and EV1B; Schulze *et al*, 2009; Magraner-Pardo *et al*, 2014; Weiner *et al*, 2015; Sadeh *et al*, 2016). In contrast, H3K79me1 was found in transcribed as well as intergenic regions (Figs 1D and EV1B). This is consistent with published ChIP-seq data and our previous ChIP-qPCR results (Weiner *et al*, 2015; Vlaming *et al*, 2016). In agreement with the distributive mechanism of methylation of Dot1 (Frederiks *et al*, 2008; De Vos *et al*, 2011), H3K79me1 and H3K79me3 anti-correlated, and H3K79me1 over the gene body was found on the minority of genes that lacked H3K79me3 and H2Bub (Fig 1D). Among these low H3K79me3, high H3K79me1 genes were subtelomeric genes, where the SIR silencing

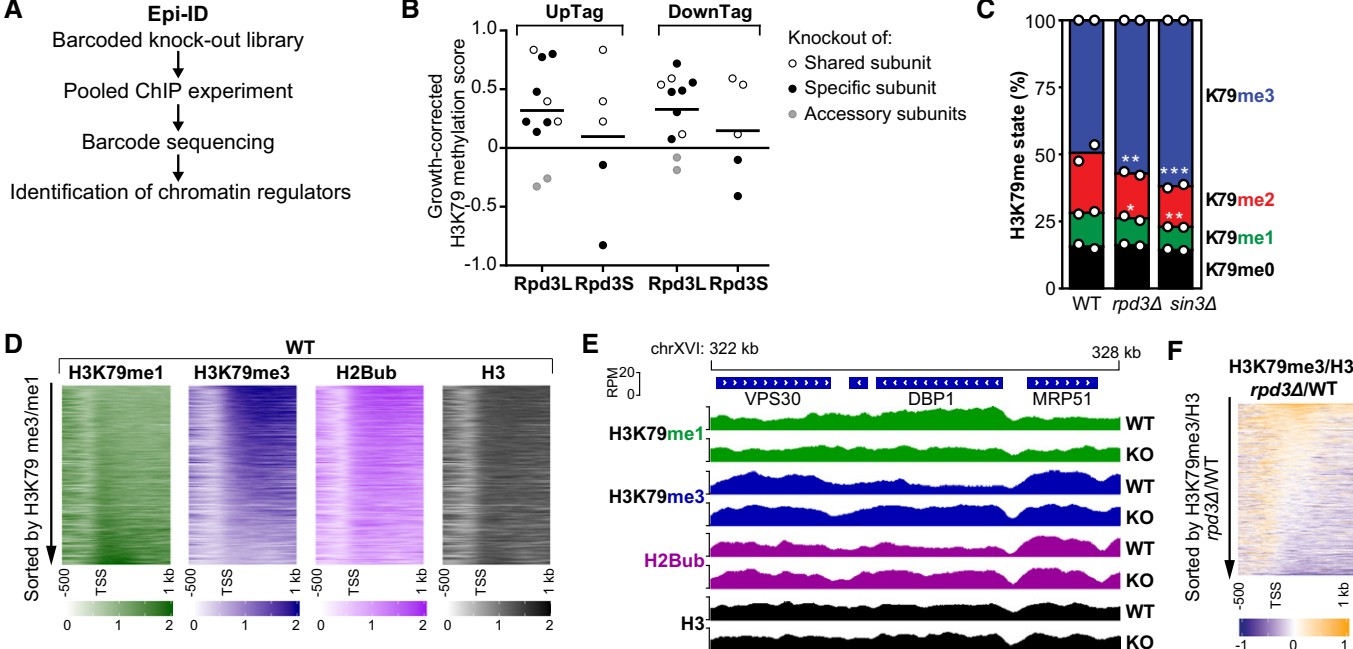

**Figure 1. Rpd3 and other members of the Rpd3L complex negatively regulate H3K79 methylation.**

A   Schematic overview of the Epi-ID strategy.

B   Epi-ID H3K79 methylation scores of the deletion mutants of members of the Rpd3L and Rpd3S complexes, calculated as described in Appendix Supplementary Methods, where 0 means a wild-type H3K79me level (log2 scale). The gray dots indicate accessory subunits. UpTag and DownTag are barcode reporters in a promoter and terminator context, respectively. Data were obtained on all Rpd3L/Rpd3S subunits except Sds3.

C   Mass spectrometry analysis of H3K79 methylation in wild-type and mutant strains. Mean and individual data points of two biological replicates. *P < 0.05, **P < 0.01, and ***P < 0.001 by two-way ANOVA, comparison to wild type.

D   Heatmaps of H3K79me1, H3K79me3, H2Bub, and H3 in wild-type cells, aligned on the TSS. Genes were sorted based on the average H3K79me3/H3K79me1 ratio in the first 500 bp.

E   Snapshot of depth-normalized ChIP-seq data tracks from wild-type and *rpd3Δ* strains showing 6 kb surrounding the *DBP1* ORF, which is the top gene in the heatmap in panel (F). All tracks have the same y-axis (0–20 rpm). A snapshot of another top-regulated gene is shown in Fig EV1D.

F   Heatmap of the H3K79me3/H3 change in *rpd3Δ* versus wild-type cells, aligned on the TSS. Genes were sorted based on the average ratio in the first 500 bp.

Source data are available online for this figure.

complex competes with Dot1 for binding to nucleosomes and H2Bub levels are kept low by the deubiquitinating enzyme Ubp10 (Gardner *et al*, 2005; Emre *et al*, 2005; Gartenberg & Smith, 2016; Kueng *et al*, 2013; Fig EV1C and E).

We then compared the patterns in wild-type versus *rpd3Δ* mutant strains. In metagene plots, the mutant showed a decrease in H3K79me1 and an increase in H3K79me3 just after the transcription start site (TSS; Fig EV1B), suggesting that in this region Rpd3 suppresses the transition from lower to higher H3K79me states. To assess whether the changes observed in the metagene plots were explained by a modest effect on H3K79me at all genes or a stronger effect at a subset of genes, we determined the H3-normalized H3K79me3 level in the first 500 bp of each gene and ranked the genes based on the change in H3K79me3 upon loss of Rpd3. A heatmap of H3K79me3 changes by this ranking showed that the absence of Rpd3 leads to an increase in H3K79me3 at a subset of genes (Fig 1F).

## Rpd3 represses H3K79me at its target genes

To characterize the genes at which H3K79me is regulated, we calculated the levels of H3K79me1 and H3K79me3 per gene in the same

500-bp window and plotted values in the rank order of H3K79me3 changes described above, using locally weighted regression (Fig 2A; corresponding heatmaps can be found in Fig EV2A). Inspection of these plots revealed that the ORFs on which H3K79me3 was increased in the *rpd3Δ* mutant showed a simultaneous decrease in H3K79me1 (groups III–IV; Fig 2A). Strikingly, these Rpd3-regulated ORFs were on average marked with a relatively high level of H3K79me1 and low H3K79me3 in the wild-type strains but became more similar to the average yeast gene upon loss of Rpd3, consistent with the presence of a negative regulator of H3K79me acting on these ORFs. Next, we compared the genes with H3K79me changes with published data on Rpd3 binding and H4 acetylation (McKnight *et al*, 2015; Data ref: McKnight *et al*, 2015). The genes with the strongest increase in H3K79me3 upon Rpd3 loss had the highest Rpd3 occupancy, both at the promoter and in the 500-bp window downstream of the TSS (group IV; Fig 2A). Rpd3 was also found to be active at these genes, since they were devoid of H4 acetylation in wild-type cells and H4 acetylation was restored in the *rpd3Δ* mutant (Fig 2A). The role of the deacetylase activity of Rpd3 was confirmed by ChIP-qPCR analysis of two previously characterized mutants of Rpd3 that lack catalytic activity (Kadosh & Struhl, 1998; Sun &

Hampsey, 1999). While re-expression of wild-type Rpd3 in the *rpd3Δ* strain restored low H3K79me3 levels at Rpd3 target genes, the Rpd3-H188A and Rpd3-H150A-H151A mutants did not rescue the loss of *RPD3* (Fig 2B). Therefore, we conclude that the Rpd3 controls H3K79 methylation via its deacetylase activity. Finally, the top-regulated genes were also enriched for meiotic genes, which are known as Rpd3 targets, and binding sites of Ume6, the Rpd3L subunit known to recruit Rpd3 to early meiotic genes (Fig 2C and D) (Kadosh & Struhl, 1998; Rundlett *et al*, 1998; Carrozza *et al*, 2005; Lardenois *et al*, 2015). Together, our results suggest that the genes at which Rpd3 restricts the buildup of H3K79me are direct targets of Rpd3.

Notably, a small subset of genes loses H3K79me3 in the absence of Rpd3 (Fig 1F, group I in Fig 2A). This group of genes already has low H3K79me3 and H2Bub levels in wild-type cells and is highly

enriched for subtelomeric genes (Fig 2A and E). Loss of Rpd3 is known to enhance Sir-mediated silencing at subtelomeric regions (Ehrentraut *et al*, 2010, 2011; Gartenberg & Smith, 2016; Thurtle-Schmidt *et al*, 2016). Our findings show that the stronger transcriptional silencing occurs with a concomitant reduction in H3K79me3 and H2Bub in the coding regions of heterochromatic genes. Whether or not the loss of these modifications contributes to the stronger silencing in *rpd3Δ/sin3Δ* mutants or is a consequence of it remains to be determined.

### Strong repression of H3K79me by Rpd3 coincides with repression of H2Bub and transcription

To understand the mechanistic basis for the crosstalk between Rpd3 and Dot1, we looked into other known functions of Rpd3 and other

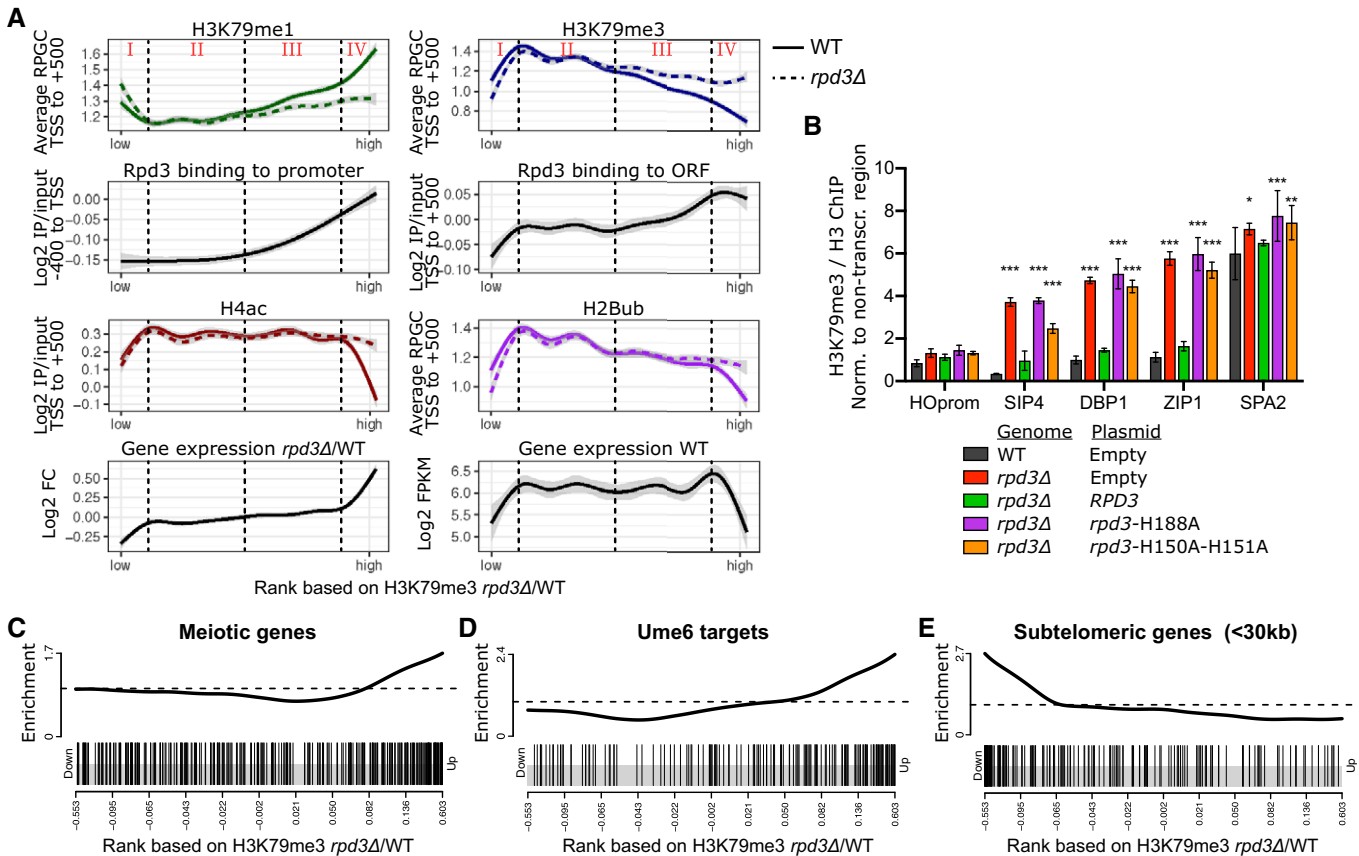

**Figure 2. Rpd3 represses transcription, H2B ubiquitination, and H3K79 methylation at its target sites.**

A   ChIP-seq and RNA-seq data for genes ranked on H3K79me3/H3 in *rpd3Δ*/WT, smoothed using locally weighted regression. The gray band around the line shows the 95% confidence interval. Vertical dashed lines separate 4 groups with distinct changes upon *RPD3* deletion. ChIP-seq data of H3K79me1, H3K79me3, and H2Bub were generated in this study (plotted is the average coverage in reads per genomic content, RPGC), Rpd3 binding, H4ac, and WT gene expression data were from McKnight *et al* (2015), and the relative expression in *rpd3Δ*/WT was from Kemmeren *et al* (2014).

B   H3K79me3/H3 ChIP-qPCR efficiencies (relative to a non-transcribed region, which was unaffected by *RPD3* deletion) in wild-type and in *rpd3Δ* cells harboring empty or *RPD3*-encoding CEN plasmids. The H188A and H150A-H151A mutations have previously been shown to abrogate catalytic activity (Kadosh & Struhl, 1998). Error bars indicate standard deviation of three biological replicates. *$P < 0.05$, **$P < 0.01$, and ***$P < 0.001$ by two-way ANOVA, comparison to wild type.

C–E   Gene set enrichment analysis on genes ranked on H3K79me3/H3 in *rpd3Δ*/WT; all genes have been ranked, and the ranks of the genes in the subsets are indicated by vertical lines. Meiotic (C) and Ume6-bound (D) genes are enriched among the genes at which Rpd3 represses H3K79 methylation, and subtelomeric genes (<30 kb of telomere) (E) are enriched among genes at which H3K79 methylation is decreased in *rpd3Δ* cells.

Source data are available online for this figure.

known regulators of H3K79me. Given the role of Rpd3 in repressing antisense transcription (Venkatesh *et al*, 2013; Castelnuovo *et al*, 2014; Murray *et al*, 2015), we compared our H3K79me data with data on antisense transcription in wild-type cells (as calculated by Brown *et al* (2018) using data from Churchman and Weissman (2011)). This analysis showed that Rpd3 does not specifically affect H3K79 methylation at genes with high or low antisense transcription, which agrees with the notion that Rpd3 represses antisense transcription via the Rpd3S complex (Venkatesh *et al*, 2013; Castelnuovo *et al*, 2014; Murray *et al*, 2015) while it regulates H3K79me via the Rpd3L complex (Fig 1B). We also compared the changes in H3K79 methylation in mutants lacking Rpd3 with changes in H3K79 methylation in mutants lacking INO80 (Data ref: Xue *et al*, 2015). This chromatin remodeler has been shown to keep transcription and H3K79me3 at intergenic regions at bay (Xue *et al*, 2015, 2017). Our analysis shows that inactivation of the INO80 complex by deletion of *ARP5* does not specifically affect H3K79 methylation at the promoters of genes that are also regulated by Rpd3, although INO80 might affect Rpd3 target genes somewhat more than non-target genes (Fig EV2F). This finding is in agreement with the observation that INO80 affects the majority of genes in the yeast genomes (Xue *et al*, 2015, 2017) whereas Rpd3 regulates H3K79 methylation at a smaller subset of genes. Finally, we examined the role of H2B ubiquitination in mediating the effect of Rpd3. The expression of the H2Bub machinery is not deregulated in Rpd3L mutants (Kemmeren *et al*, 2014), and no upregulation of H2Bub could be detected by immunoblot (Fig EV1A). Because subtle changes can be missed by blot, we proceeded to generate H2Bub ChIP-seq data in wild-type and mutant strains using an antibody we recently described (Van Welsem *et al*, 2018). We found that the strongest H3K79me3 repression by Rpd3 (group IV) coincided with repression of H2Bub as well as transcription (Fig 2A; RNA-seq data from McKnight *et al* (2015); Data ref: McKnight *et al*, 2015). Moreover, these genes had below-average H2Bub and transcription levels in wild-type cells (Fig 2A). H2Bub changes were confirmed by ChIP-qPCR (Fig EV2B and C).

The H2B ubiquitination machinery is known to be recruited co-transcriptionally and promote H3K79 methylation, so transcriptional repression provides a likely explanation for the restriction of H3K79 methylation at these genes. However, despite these established causal links, there is no simple linear relation between transcription level and H3K79me3 level, while H2Bub correlates with transcription perfectly (Fig EV2D; Schulze *et al*, 2009, 2011; Weiner *et al*, 2015). It appears that other processes counteract H3K79 methylation (see Discussion), especially at highly transcribed genes, but that these processes do not affect the Rpd3-regulated genes as much, since they form a subset of genes at which transcription and H3K79me3, and their changes upon *RPD3* deletion, are correlated.

In addition to genes where Rpd3 has a strong effect on H3K79me, we also observed genes at which H3K79 methylation was more modestly affected by the deletion of *RPD3* (group III; Fig 2A). Rpd3 is found at the promoters of these genes, but H4 acetylation, transcription, and H2B ubiquitination are not affected (Fig 2A). Although some of the differences may be caused by differences in antibody strength, the results suggest that at these loci, another, still unknown additional mechanism could be at play.

Taken together, we identified Rpd3 as a bona fide negative regulator of H3K79 methylation in yeast that restricts H3K79me3 at its euchromatic targets, probably mostly by repressing target gene

transcription and H2Bub, but other mechanisms seem to be involved as well.

### HDAC1 loss increases H3K79me in murine thymocytes

Having uncovered a role for Rpd3 in restricting H3K79me at its targets and finding that this can explain a significant part of the H3K79 methylation variance between genes in yeast, we next wanted to investigate the biological relevance of this regulation in mammals. Histone deacetylases are conserved between species and can be divided into four classes (Yang & Seto, 2008). Rpd3 is a founding member of the class I HDACs, which in mammals includes HDAC1, HDAC2, HDAC3, and HDAC8. Of these, HDAC1 and HDAC2 are found in Sin3 complexes, like yeast Rpd3 (Yang & Seto, 2008). Given that both HDACs and DOT1L play critical roles in T-cell malignancies, we employed conditional early thymocyte-specific *Hdac1* deletion (*Lck*-Cre;*Hdac1*$^{f/f}$, resulting in *Hdac1*$^{\Delta/\Delta}$ thymocytes) in the mouse to investigate whether the regulation that we observed in yeast also exists in T cells. We focused on HDAC1 because it is more active in mouse thymocytes than HDAC2 (Dovey *et al*, 2013; Heideman *et al*, 2013). First, we measured the relative abundance of H3K79 methylation states on bulk histones by mass spectrometry in wild-type and *Hdac1*-deleted thymocytes of 3-week-old mice (Fig 3A). In general, the overall levels of H3K79 methylation were much lower than in yeast and H3K79me1 was the most abundant methylation state, followed by H3K79me2, consistent with previous reports in mouse and human cells (Jones *et al*, 2008; Leroy *et al*, 2013). As seen in Fig 3A, *Hdac1*-deleted thymocytes had more H3K79me2 and H3K79me1 and less H3K79me0. Considering the distributive activity of Dot1 enzymes (Fig EV3A), this suggests that Rpd3/HDAC1 is a conserved negative regulator of H3K79 methylation.

### Reduced DOT1L dosage increases the latency of *Hdac1*-deficient thymic lymphomas

Conditional *Lck*-Cre;*Hdac1*$^{f/f}$ knock-out mice die of thymic lymphomas characterized by loss of p53 activity and *Myc* amplification, with a 75% incidence and a 23-week mean latency (Heideman *et al*, 2013). Oncogenic transformation has not occurred yet in 3-week-old mice (Heideman *et al*, 2013), the age at which H3K79me levels were determined above. Since *Hdac1* deletion in thymocytes resulted in an increase in H3K79 methylation, as well as thymic lymphoma formation, we asked whether increased H3K79 methylation was important for tumor development in this mouse model. To address this question, a conditional *Dot1L* (*Dot1L*$^{f/f}$) allele was crossed into the *Lck*-Cre;*Hdac1*$^{f/f}$ line such that deletion of *Hdac1* was combined with deletion of zero, one or two *Dot1L* alleles. Immunohistochemistry confirmed the loss of HDAC1 at the protein level, and mass spectrometry confirmed the loss of DOT1L protein activity for the expected genotypes (Figs 3A and B, and EV3B). H3K79me2 was used as an indicator for DOT1L presence, since none of the DOT1L antibodies we tested worked for IHC (antibody difficulties have also been described by Sabra *et al*, 2013).

Mice with conditional *Hdac1* alleles but wild type for *Dot1L* (*Lck*-Cre;*Hdac1*$^{f/f}$) developed thymic lymphomas for which they had to be sacrificed, with a median latency of 21 weeks and an incidence of 86% during the 40-week length of the experiment, comparable to

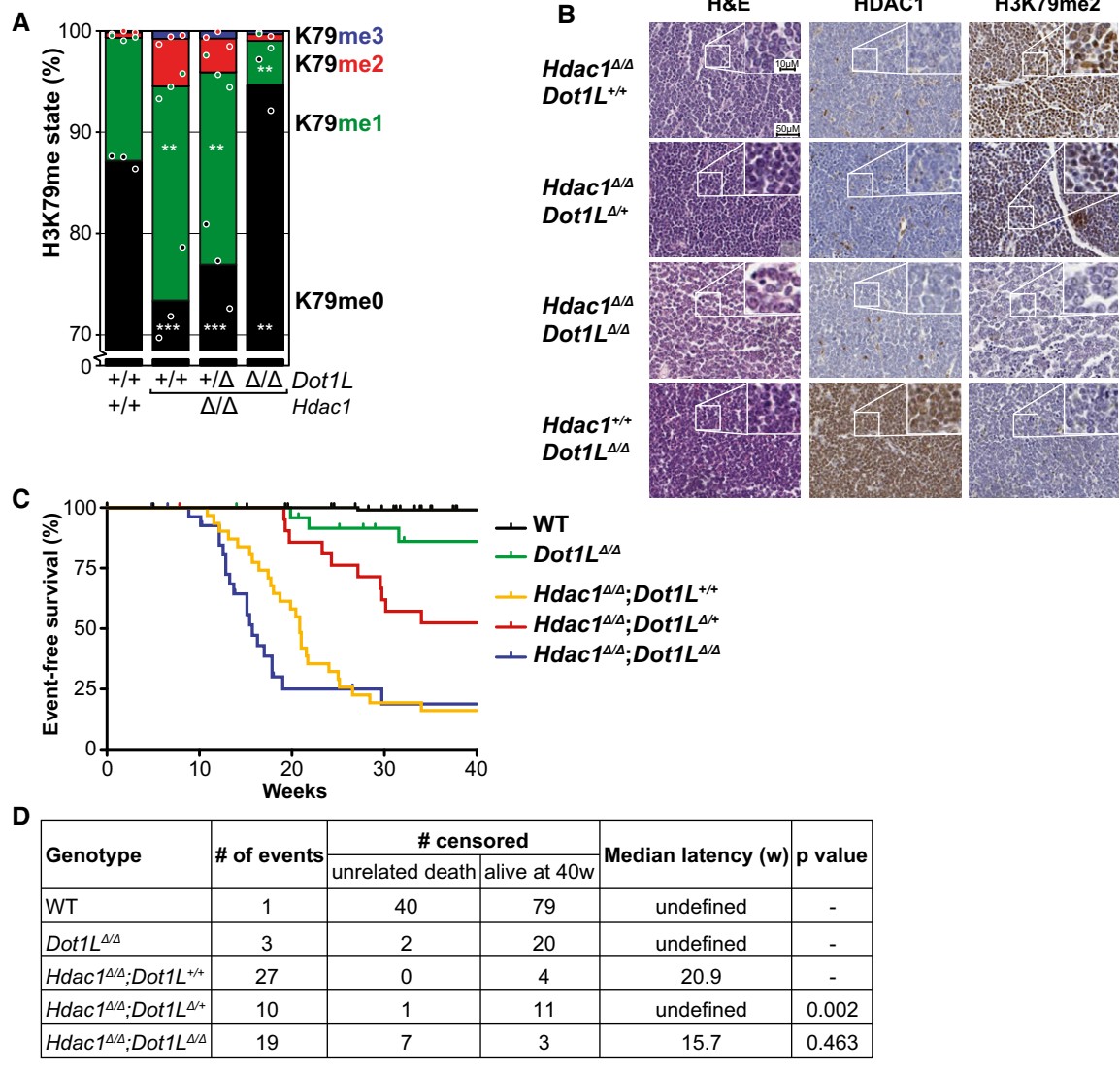

**Figure 3. *Hdac1* deletion increases H3K79 methylation in thymocytes *in vivo*, and simultaneous heterozygous *Dot1L* deletion prolongs tumor-free survival.**

A   Mass spectrometry analysis of H3K79 methylation in thymuses from 3-week-old mice, either wild-type (Cre-) or with deleted *Hdac1* or *Dot1L* alleles, as indicated. Mean and individual data points of biological replicates; H3K79me0 is the predominant state, and the y-axis is truncated at 70% for readability. The remaining H3K79 methylation after homozygous *Dot1L* deletion is probably due to the presence of some cells in which Cre is not expressed (yet). \*\**P* < 0.01 and \*\*\**P* < 0.001 by two-way ANOVA, comparison to wild type.

B   Representative H&E and immunohistochemical staining on sequential sections of thymic lymphomas of the indicated genotypes. A picture with lower magnification of independent samples is included in Fig EV3B.

C   Kaplan–Meier curves of mice harboring thymocytes with indicated genotypes. An event was defined as death or sacrifice of a mouse caused by a thymic lymphoma. Mice that died due to other causes or were still alive and event-free at the end of the experiment were censored. Mice for which the cause of death could not be determined were removed from the data. Wild-type mice were the Cre- littermates of the mice that were used for the other curves.

D   Summary of the data presented in panel C. A median latency could only be calculated when the tumor incidence was > 50%. The *P* value was determined by comparing to the *Lck*-Cre;*Hdac1*^f/f curve with a Peto test, but a logrank test yielded the same conclusions.

Source data are available online for this figure.

what was observed before (Fig 3C and D) (Heideman *et al*, 2013). As expected, *Lck*-Cre-negative control mice rarely developed thymic lymphomas (1 out of 112). Also, *Dot1L* deletion alone (*Lck*-Cre; *Dot1L*^f/f) rarely led to thymic lymphomas, with a 15% incidence in this background (Fig 3C and D), and no cases of thymic lymphoma in another background (data not shown). We then assessed the effect of *Dot1L* deletion in the *Lck*-Cre;*Hdac1*^f/f model. Loss of one

copy of *Dot1L* increased survival rate and tumor latency (48% incidence, comparison to *Hdac1*^f/f alone *P* = 0.002; Fig 3C and D). This effect suggests that there is a causal link between the increase in H3K79 methylation and the development or maintenance of thymic lymphomas upon *Hdac1* deletion. Interestingly, homozygous *Dot1L* deletion, leading to a complete loss of H3K79me, did not extend the latency of thymic lymphomas (81% incidence, 15.7-week median

latency, comparison to *Hdac1* deletion alone *P* = 0.463; Fig 3C and D). A possible explanation is that the simultaneous deletion of *Dot1L* and *Hdac1* results in the generation of a different class of tumor that does not depend on H3K79 methylation but has acquired other, possibly epigenetic, events that allow oncogenic transformation. A similar model has been proposed for the loss of *Hdac2* in the *Lck*-Cre;*Hdac1*^f/f^ model (Heideman *et al*, 2013). For the heterozygous *Dot1L* effect, we consider two possible explanations: The oncogenic transformation occurred later because an additional event was required to overcome the lack of high H3K79me, or tumors grew slower due to lower H3K79me, because of either decreased proliferation or increased apoptosis.

## *Hdac1*-deficient thymic lymphoma lines depend on DOT1L activity

To further study the *Dot1L* dependence of *Hdac1*-deficient thymic lymphomas in a more controlled environment, we turned to *ex vivo* experiments. Cell lines were derived from *Hdac1*-deficient thymic lymphomas (Heideman *et al*, 2013). Since these cell lines had an inactivating mutation in p53, cell lines derived from p53-null thymic lymphomas were used as *Hdac1*-proficient control lines (Heideman *et al*, 2013). First, we examined whether *Hdac1*-deficient tumor cells retained the increased H3K79 methylation levels seen prior to the oncogenic transformation. Both by immunoblot and by targeted mass spectrometry on independent samples (Fig 4A and B), *Hdac1*-deficient tumor cell lines had more H3K79 methylation than their *Hdac1*-proficient counterparts. Thus, the effect of HDAC1 on H3K79 methylation observed *in vivo* in 3-week-old pre-malignant thymuses was maintained in the thymic lymphoma cell lines. Importantly, *Hdac1*-deficient cell lines also possessed high levels of ubiquitinated H2B compared to *Hdac1*-proficient controls (Fig 4C). High H2Bub is consistent with the increase in H2Bub seen at Rpd3 targets in yeast. Therefore, a plausible model is that at least part of the observed increase in H3K79me upon loss of HDAC1 activity is mediated via H2Bub, consistent with what we observed in yeast. However, contributions from other regulatory mechanisms cannot be excluded (see Discussion). To test the DOT1L dependence of the cell lines, DOT1L was depleted using shRNAs in *Hdac1*-proficient and *Hdac1*-deficient cell lines. As can be seen in Fig 4D, shRNAs that reduce Dot1L expression (Fig EV4A) affected proliferation of the *Hdac1*-deficient cell lines. Compared to the control lines, the *Hdac1*-deficient cell lines were also more sensitive to two different DOT1L inhibitors (Fig 4E). Both inhibitors, EPZ-5676 (Pinometostat) and SGC-0946, effectively lowered H3K79 methylation (Fig EV4B). Thus, shRNA-mediated DOT1L knockdown and chemical DOT1L inhibition showed that the *Hdac1*-deficient thymic lymphoma cell lines depended on DOT1L activity.

The reduced growth upon inactivation of DOT1L could be explained by a block in proliferation or an increase in cell death. To measure apoptosis induction, the levels of Annexin V and DAPI staining of non-permeabilized cells were determined by flow cytometry. In the DMSO-treated condition, most cells were alive, although *Hdac1*-deficient lines had a slightly higher basal apoptosis level (Fig 5A and B). This combination of proliferation and apoptosis has also been observed in *Hdac1*^Δ/Δ^ teratomas (Lagger *et al*, 2010). However, DOT1L inhibition by 5µM of SGC-0946 dramatically induced apoptosis in *Hdac1*-deficient cells, whereas no effect on

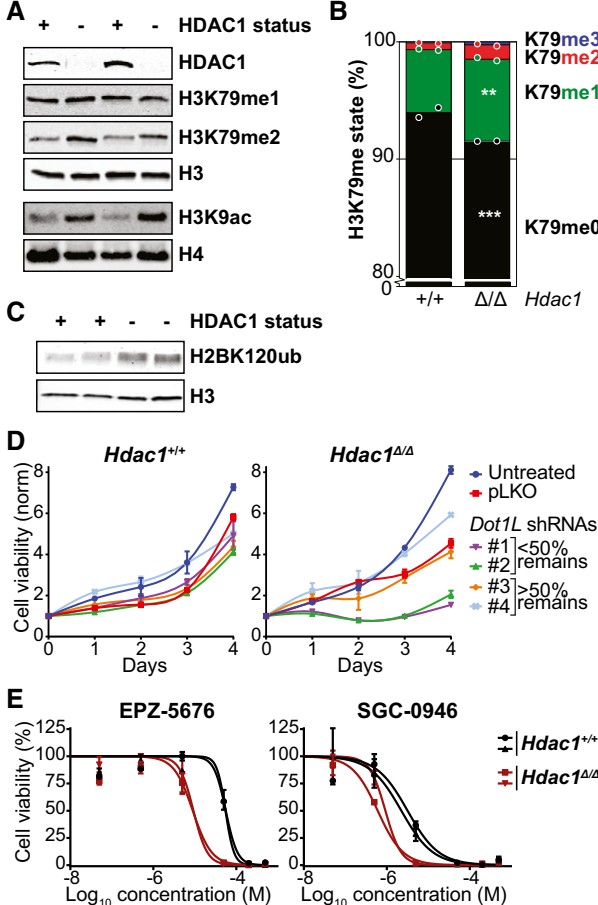

**Figure 4. *Hdac1*-deleted thymic lymphoma cell lines depend on DOT1L activity.**

A Immunoblots showing HDAC1 status and H3K79me/H3K9ac levels in nuclear lysates of *Hdac1*-proficient (p53-null) and *Hdac1*-deficient thymic lymphoma cell lines. The top four and bottom two panels are from separate lysates of the same cell lines.

B Mass spectrometry analysis of H3K79 methylation in the cell lines from panel A. Mean and individual data points of two independent cell lines; H3K79me0 is the predominant state, and the y-axis is truncated at 80% for readability. **P < 0.01 and ***P < 0.001 by two-way ANOVA.

C Immunoblot showing H2BK120 ubiquitination levels in Hdac1-proficient and Hdac1-deficient cell lines (two independent lines each).

D Growth curves of *Hdac1*-proficient and *Hdac1*-deficient cell lines that were left untreated or were infected with empty virus (pLKO) or shRNAs against *Dot1L* and selected with puromycin. Growth curves were determined by a series of resazurin assays, which measure metabolic activity, starting from four days post-infection. Error bars indicate the range of two replicates from independent cell lines.

E Inhibitor dose–response curves of the two DOT1L inhibitors EPZ-5676 (Pinometostat) and SGC-0946 in *Hdac1*-proficient and *Hdac1*-deficient cell lines. Cell viability was measured by a resazurin assay after three days of treatment, and measurements were normalized to DMSO-treated cells. Two independent cell lines are plotted separately; error bars indicate the range of two biological replicates.

Source data are available online for this figure.

apoptosis was observed in the control cell lines (Fig 5A and B). Thus, DOT1L inhibition induced apoptosis specifically in *Hdac1*-deficient thymic lymphoma cell lines.

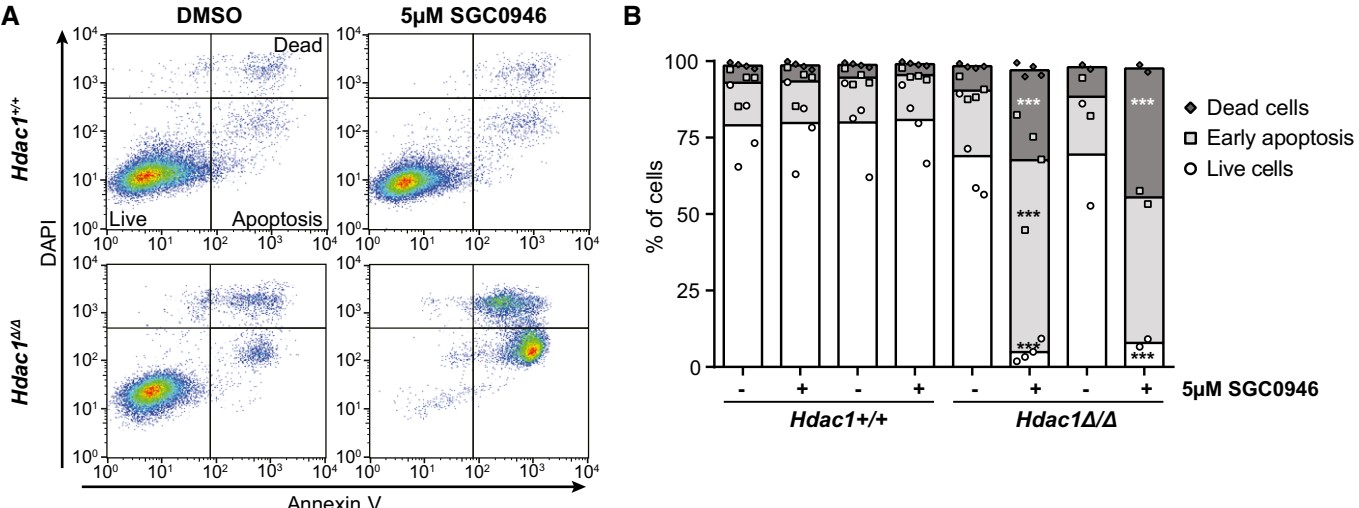

**Figure 5. *Hdac1*-deficient thymic lymphoma lines require DOT1L activity for survival.**

A Representative apoptosis FACS plots of cell lines treated with DMSO or the DOT1L inhibitor SGC-0946 for two days. Annexin V staining and DAPI staining were performed on unpermeabilized cells to distinguish live (Annexin V low; DAPI low), apoptotic (Annexin V high; DAPI low), and dead (Annexin V high; DAPI high) cells.

B Quantification of several independent FACS experiments, including the experiment shown in panel (A). Mean with individual data points of 2–4 replicates each of two independent lines per genotype. ***$P < 0.001$ by two-way ANOVA, comparison to corresponding DMSO control.

Source data are available online for this figure.

## Discussion

Here, we describe that the yeast HDAC, Rpd3, is a negative regulator of H3K79 methylation that restricts methylation at the 5′ ends of its target genes. Similar to what we observe for Rpd3 in yeast, deleting *Hdac1* in murine thymocytes leads to an increase in H3K79 methylation. This regulation is relevant in a cancer context, since heterozygous deletion of *Dot1L* prolongs the survival of mice that develop *Hdac1*-deficient thymic lymphomas. Cell lines derived from *Hdac1*-deficient thymic lymphomas undergo apoptosis upon DOT1L inhibition or depletion, which indicates a form of non-oncogene addiction to DOT1L.

### Rpd3 target genes

In the yeast genome, most euchromatic genes are marked by H2Bub and H3K79me3 in their transcribed region. While the levels of H2Bub correlate well with transcription levels, it is evident that H2Bub is not the only determinant of H3K79me3 in yeast because the relation between transcription and H3K79me3 is more complex (Fig 2A). While genes silenced by the SIR complex have low H3K79me3 levels due to active repression mechanisms (Gartenberg & Smith, 2016), the majority of euchromatic genes contain H3K79me3 irrespective of their expression level (Fig EV2D; Schulze *et al*, 2009, 2011; Weiner *et al*, 2015). Some genes contain lower H3K79me3 and higher H3K79me1 levels than the average gene, however. A subset of these deviants has been identified as genes undergoing antisense transcription (Murray *et al*, 2015; Brown *et al*, 2018), possibly resulting in nucleosome instability and increased histone turnover, which counteracts the buildup of higher H3K79me states but does not affect the more dynamic H2Bub modification

(Weiner *et al*, 2015). Here, we provide insight into the low H3K79me3/high H3K79me1 levels of another subset of yeast genes. H3K79me ChIP-seq in yeast revealed that Rpd3 restricts H3K79me3 at its direct target genes. This subset of genes showed great overlap with the minority of euchromatic genes that is marked with H3K79me1 instead of H3K79me3. Thus, the regulation by Rpd3 provides an explanation for the variation in H3K79 methylation between genes and thereby seems to be an important determinant of the H3K79 methylation pattern. The H3K79me effect of Rpd3 was most notable at the 5′ end of genes, which is in agreement with previous studies on Rpd3 activity. The deacetylation activity of Rpd3/Rpd3L is reported to be strongest in coding sequences, particularly at the 5′ ends (Weinberger *et al*, 2012). At which genes HDAC1 regulates H3K79 methylation in murine thymocytes is an interesting question, but addressing it is not straightforward. Unlike in yeast, in mammals H3K79 methylation is tightly linked to the transcriptional activity at genes. Processes through which transcription promotes H3K79 methylation are known, but in turn, H3K79 methylation may affect transcription as well (reviewed in Vlaming & Van Leeuwen, 2016). Therefore, when assessing H3K79me changes in *Hdac1*-deficient cells, it will be challenging to separate direct effects from indirect effects on DOT1L activity through transcriptional changes.

### Mechanism of regulation

What could be the mechanism of H3K79me regulation by Rpd3/HDAC1? Until now, the H2B ubiquitination machinery was the only described H3K79me regulator conserved from yeast to mammals (Weake & Workman, 2008). Here, we describe another conserved regulator, Rpd3/HDAC1, and our results indicate that it acts in part

by restricting H2Bub. Although transcription and H3K79 methylation are not clearly positively correlated in wild-type yeast cells (Fig EV2D; Schulze *et al*, 2011; Magraner-Pardo *et al*, 2014; Weiner *et al*, 2015), we observed that the H3K79 methylation changes in *rpd3Δ* correlate very well with transcriptional changes. These data, together with the well-established causal relationships between transcription and H2B ubiquitination on the one hand and H2B ubiquitination and H3K79 methylation on the other hand, suggest that there is indeed a causal link between (sense) transcription and the placement of H3K79 methylation at a subset of the yeast genome. At higher transcription levels however, this relationship can be obscured by other processes, most likely histone turnover, counteracting the high Dot1 activity (Radman-Livaja *et al*, 2011; Murray *et al*, 2015). We have recently identified the conserved histone acetyltransferase Gcn5 as a negative regulator of H3K79me and H2Bub (Vlaming *et al*, 2016). At first glance, it seems counterintuitive that a HAT and an HDAC have overlapping effects. However, acetylation at non-overlapping histone or non-histone lysines may explain this discrepancy. For example, Gcn5 most likely negatively regulates H2Bub and H3K79me by affecting the deubiquitinating module of the SAGA co-activator complex in which Gcn5 also resides (Vlaming *et al*, 2016). H2B ubiquitination by human RNF20/40 has also been shown to be regulated by histone acetylation (Garrido Castro *et al*, 2018). Treatment of acute lymphoblastic leukemia cell lines with the non-selective HDAC inhibitor Panobinostat showed changes in H2Bub, with decreased H2Bub in MLL-r leukemia lines and increased H2Bub in non-MLL-r leukemia lines, suggesting context-dependent mechanisms (Garrido Castro *et al*, 2018).

Besides H2Bub, other mechanisms are likely to contribute to the observed H3K79me increase in the absence of Rpd3/HDAC1 as well. Histone acetylation is increased in the absence of the deacetylase HDAC1, and histone acetylation has been previously linked to DOT1L recruitment through the YEATS domain transcription elongation proteins AF9 and ENL (Li *et al*, 2014; Kuntimaddi *et al*, 2015; Erb *et al*, 2017; Wan *et al*, 2017). Very recently, preferential Dot1 binding to acetylated H4K16 has been shown, and the histone acetyltransferase Sas2 was found to be a positive regulator of H3K79 methylation in yeast, probably via acetylation of H4K16 (Lee *et al*, 2018). Our identification of Rpd3/HDAC1 as a regulator of DOT1L underscores the intimate relationship between histone acetylation and H2Bub and H3K79me and provides evidence for a specific HDAC involved in the crosstalk: HDAC1.

### DOT1L in tumor maintenance

Loss of HDAC1 leads to oncogenic transformation and higher H3K79me in murine thymocytes. Heterozygous *Dot1L* deletion prolonged the survival of mice with thymocyte-specific *Hdac1* deletion due to a lower incidence and increased latency of thymic lymphomas. Our analysis of *Hdac1*-deficient thymic lymphoma cells *ex vivo* provided more insight into the possible mechanisms for the reduced tumor burden. Using DOT1L inhibitors and a knockdown approach, we established that DOT1L was required for survival of the tumor cells by preventing the induction of apoptosis, suggesting that DOT1L is required for tumor maintenance. The DOT1L dependency of the thymic lymphomas resembles that of MLL-rearranged leukemia (Wang *et al*, 2016) as well as breast and lung cancer cell lines (Kim *et al*, 2012; Zhang *et al*, 2014). The full genetic deletion of *Dot1L* did not reduce tumor burden. The reasons for this are currently unknown and require further study. One possible reason is that some remaining DOT1L activity and H3K79 methylation might be required to induce apoptosis in the tumor cells. We note that in the *ex vivo* experiments where *Dot1L* knockdown and DOT1L inhibitors were found to lead to induction of apoptosis; some residual H3K79me was indeed still present. Another possibility is that the simultaneous loss of *Hdac1* and *Dot1L* imposes oncogenic transformation through alternative, epigenetic mechanisms that bypass the apoptotic-prone state. This would be in agreement with the known role of DOT1L in the maintenance of cellular epigenetic states (Onder *et al*, 2012; Soria-Valles *et al*, 2015; Breindel *et al*, 2017). Regardless of possible mechanisms, the finding that HDAC1 affects DOT1L activity in yeast as well as mouse T cells warrants further investigation. For example, it will be interesting to determine whether and under which conditions HDAC1 activity influences DOT1L activity in human MLL-r leukemia and whether the crosstalk is involved in the response of CTLC to HDAC inhibitors in the clinic. Our findings in murine lymphoma add to a growing list of cancers that rely on DOT1L activity, and therefore underline the importance of understanding the regulation of DOT1L.

## Materials and Methods

### Yeast strains and plasmids

Yeast strains used in this article are listed in Appendix Table S1. Yeast media were described previously (Van Leeuwen & Gottschling, 2002). The generation of the barcoded deletion library used for the Epi-ID experiment was described previously (Vlaming *et al*, 2016). Yeast *rpd3Δ* and *sin3Δ* strains were taken from this library and independent clones were generated by deleting these genes in the barcoded wild-type strain NKI4657, using the NatMX selection marker from pFvL99 (Stulemeijer *et al*, 2011). Gene deletions were confirmed by PCR. *RPD3* expression vectors were derived from YCplac22-RPD3, YCplac22-rpd3_H188A, and YCplac112-rpd3_H150A_H151A (Kadosh & Struhl, 1998). The inserts were released by digestion with *Sac*I and *Sal*I and cloned into the same sites of the single-copy *LEU2* vector pRS315 (Brachmann *et al*, 1998). The mutations were verified by DNA sequencing. Strains harboring pRS315-derived plasmids were grown in synthetic media lacking leucine.

### Cell culture

Thymic lymphoma cell lines (NKI8996, NKI9002) were derived from *Hdac1*-deficient thymic lymphomas (Heideman *et al*, 2013). Since these cell lines had an inactivating mutation in p53, cell lines derived from p53-null thymic lymphomas (NKI8995, NKI8999) were used as *Hdac1*-proficient control lines (Heideman *et al*, 2013). Thymic lymphoma cell lines were cultured under standard conditions in RPMI 1640 (Gibco) supplemented with 10% FBS (Sigma-Aldrich), antibiotics, and L-glutamine. The HDAC1 status was confirmed by immunoblot analyses (Fig 4A). HEK 293T cells were cultured in DMEM (Gibco) supplemented with 10% FBS

(Sigma-Aldrich) and L-glutamine. Cell lines were regularly tested for mycoplasma contamination.

## Mouse survival analysis

The generation and crosses of the conditional knock-out mice are described in the Appendix Supplementary Methods. Mice were monitored over time, up to 40 weeks of age. A power analysis performed beforehand determined group sizes of 20–25 mice per genotype. Mice were sacrificed before 40 weeks when they displayed breathing issues caused by the thymic lymphoma, or serious discomfort unrelated to tumor formation. After death or sacrifice, mice were checked for the presence of a thymic lymphoma. Mice that died without a lymphoma were censored in the survival curves. Mice of which the cause of death could not be determined were left out of the survival curve, together with their littermates. Mice were kept in individually ventilated cages at the animal laboratory facility of the Netherlands Cancer Institute (NKI; Amsterdam, the Netherlands). Food and water were provided ad libitum. Animal experiments were approved by the Animal Ethics Committee of the NKI and performed in accordance with institutional, national, and European guidelines for animal care and use. The study is compliant with all relevant ethical regulations regarding animal research.

## Protein lysates, immunoblots, and mass spectrometry

Yeast whole-cell extracts were made as described previously (Vlaming *et al*, 2014). Nuclear extracts were prepared of murine cells; see Appendix Supplementary Methods for details. The immunoblotting procedure was as described in Vlaming *et al* (2014). All antibodies used in this study are listed in the Appendix Supplementary Methods. Mass spectrometry measurements on yeast strains and thymic lymphoma cell lines were as described in Vlaming *et al* (2014). Measurements on thymus tissue were performed using the method described in Vlaming *et al* (2016).

## ChIP-sequencing and ChIP-qPCR

ChIP and ChIP-qPCR experiments were performed as described before (Vlaming *et al*, 2016). Primers for qPCR are shown in Appendix Table S2. Details on the ChIP-seq library preparation and the first analysis steps are provided in the Appendix Supplementary Methods. In short, preparation and sequencing were performed by the NKI Genomics Core Facility. Reads were mapped to the *Saccharomyces cerevisiae* reference genome R64-2-1 and extended to 150 bp. Samples were depth-normalized, and when data from biological duplicates were found to be similar, data sets were merged for further analyses. Metagene plots and heatmaps were generated with custom scripts in R/Bioconductor (Cherry *et al*, 2012; Huber *et al*, 2015). Reads were aligned in a window of −500 to 1 kb around the TSS of each verified ORF recorded in SGD. Genes that contained a coverage of 0 or an average coverage in the first 500 bp below 0.5 were filtered out (leaving 5,006 out of 5,134 genes). For heatmaps, the coverage was grouped in bins of 10 bp. The plots in Figs 2A and EV2D–F were created with custom scripts in R by using locally weighted regression (LOESS). Transcription in WT cells was obtained from McKnight *et al* (2015) and transcription

in *rpd3Δ*/WT from Kemmeren *et al* (2014). Gene set enrichment plots were created with the barcodeplot function from the limma package (Ritchie *et al*, 2015). Ume6 targets filtered for Ume6 DNA binding were obtained from the YEASTRACT database (Teixeira *et al*, 2014). The list of meiosis factors was generated by searching for genes with a GO term containing "meio" or children thereof. The distance of each gene from the telomere on the same chromosome arm was calculated manually by using genome feature information from SGD.

## Histology/immunohistochemistry

Tissues were fixed in EAF (ethanol, acetic acid, and formol saline) for 24 h and subsequently embedded in paraffin. Slides were stained with hematoxylin and eosin (H&E), or immunohistochemistry was performed as described (Heideman *et al*, 2013).

## Knockdown/viability assays and Dot1L mRNA analysis

To produce lentiviral particles, HEK 293T cells were co-transfected with shRNA-containing pLKO.1 and three packaging plasmids containing Gag and Pol, and Rev and VSV-G, respectively, using PEI. The media were replaced after 16 h, and virus-containing medium was harvested 72 h after transfection. Virus particles were 10× concentrated from filtered medium using Amicon 100 kDa spin columns. For lentiviral infections, 100,000 cells were seeded in 96-well tissue culture plates and infected using 7.5 μl concentrated virus, in the presence of 8 μg/ml polybrene. The medium of infected cells was replaced with puromycin-containing medium 48 h after infection and refreshed again 72 h after infection, after which a cell viability assay was performed every 24 h. Cell viability was determined by a CellTiter-Blue (Promega) assay, measuring conversion to resorufin after 3 h with the EnVision Multilabel Reader (PerkinElmer). All cells treated with a particular shRNA were pooled for RNA isolation using the RNeasy Mini Kit (Qiagen). DNase I (New England Biolabs) digestion was performed, and RNA was reverse-transcribed into cDNA using SuperScript II Reverse Transcriptase (Invitrogen) according to the manufacturer's protocol. *Dot1L* transcript abundance was measured by qPCR using SYBR Green Master Mix (Roche) and the LightCycler 480 II (Roche). Primers for qPCR can be found in Appendix Table S2.

## Inhibitor treatment

A total of 100,000 cells were seeded in 96-well tissue culture plates, in 200 μl culture medium containing the indicated concentration of inhibitor. Two inhibitors were used: SGC-0946 (Structural Genomics Consortium) and EPZ-5676 (Pinometostat; Selleck Chemicals). Cell viability was determined after three days, as described above. Data were normalized, with the maximum of each cell line to 100% and background fluorescence set to 0%. GraphPad Prism was used to fit log(inhibitor) vs normalized response curves with a variable slope.

## Apoptosis FACS

Cells treated with 0.1% DMSO or 5 μM SGC-0946 were stained with Annexin V-FITC and DAPI following the protocol of the Annexin V-FITC Apoptosis Detection Kit (Abcam). Fluorescence was detected

by FACS using the CyAn ADP Analyzer (Beckman Coulter), and data were analyzed using FlowJo software.

### Statistics

Survival curves were plotted in GraphPad Prism, and Peto mortality-prevalence tests were performed in SAS to compare the curve of *Lck*-Cre;*Hdac1*$^{f/f}$ mice with the *Lck*-Cre;*Hdac1*$^{f/f}$;*Dot1L*$^{f/WT}$ and *Lck*-Cre;*Hdac1*$^{f/f}$;*Dot1L*$^{f/f}$ curves. The same conclusions could be drawn based on the standard logrank test in GraphPad Prism. Mass spectrometry and ChIP-qPCR data were compared using two-way ANOVA, comparing samples of all genotypes to the wild-type sample and using the Dunnett's correction for multiple comparisons, using GraphPad Prism.

## Data availability

The ChIP-seq data from this publication have been deposited to the GEO database (www.ncbi.nlm.nih.gov/geo/) and assigned the identifier accession number GSE107331.

**Expanded View** for this article is available online.

### Acknowledgements

The authors thank Jeffrey McKnight and Toshio Tsukiyama for sharing Rpd3 and H4ac ChIP-seq data. We thank Roel Wilting and Marinus (Richard) Heideman for help with initial HDAC1 experiments, Michael Hauptman for statistical tests on the survival curves, Kevin Struhl for *RPD3* plasmids, and Struan Murray and Jane Mellor for providing source data on antisense transcription. We thank the NKI animal pathology facility for histology and immunohistochemistry, as well as advice, the NKI Genomics Core Facility for library preparations and sequencing, the NKI FACS facility for assistance, Onno Bleijerveld for mass spectrometry advice, and the caretakers of the NKI laboratory animal facility for assistance and excellent animal care. We thank Ila van Kruijsbergen, Tineke Lenstra, and Maarten van Lohuizen for critically reading the manuscript. This work was supported by the Dutch Cancer Society (KWF2009-4511 and NKI2014-7232 to FvL and HJ) and the Netherlands Organisation for Scientific Research (NWO-VICI-016.130.627, NCI-KIEM-731.013.102, and NCI-LIFT-731.015.405 to FvL and ZonMW Top 91213018 to HJ). The funders had no role in study design, data collection and interpretation, or the decision to submit the work for publication.

### Author contributions

HV, CMM, and FvL designed the studies and the cell line and mouse studies together with HJ and J-HD. Yeast experiments were performed by HV, TMM, and TvW; mouse experiments by CMM, HV, SH, EMK-M, MFA, and CL; and cell line experiments by CMM, SP, and HV. TK performed ChIP-seq and genomics analyses, SK made IHC pictures and gave histology advice, and LH and TTS performed mass spectrometry measurements and were advised by AFMA. HV and FvL wrote the manuscript, with help from CMM, HJ, and J-HD.

### Conflict of interest

The Netherlands Cancer Institute and FvL are entitled to royalties that may result from licensing the yeast H2BK123ub-specific monoclonal antibody according to IP policies of the Netherlands Cancer Institute. The other authors declare that they have no conflict of interest.

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

     Evidence that ubiquitylated H2B corrals hDot1L on the nucleosomal
     surface to induce H3K79 methylation. *Nat Commun* 7: 10589

