## [Review Process File · The EMBO Journal]

Conserved crosstalk between histone deacetylation and H3K79 methylation generates DOT1L-dose dependency in HDAC1-deficient thymic lymphoma

Hanneke Vlaming, Chelsea M. McLean, Tessa Korthout, Mir Farshid Alemdehy, Sjoerd Hendriks, Cesare Lancini, Sander Palit, Sjoerd Klarenbeek, Eliza Mari Kwesi-Maliepaard, Thom M. Molenaar, Liesbeth Hoekman, Thierry T. Schmidlin, A.F. Maarten Altelaar, Tibor van Welsem, Jan-Hermen Dannenberg, Heinz Jacobs and Fred van Leeuwen.

Review timeline:

Submission date:	15 th January 2019
Editorial Decision:	6 th March 2019
Revision received:	23 rd April 2019
Editorial Decision:	7 th May 2019
Revision received:	20 th May 2019
Accepted:	24 th May 2019

Editor: Stefanie Boehm

Transaction Report:

1st Editorial Decision

6th March 2019

Thank you for submitting your manuscript for consideration by The EMBO Journal and I apologize again for the delay of the reviewing process. We had been waiting for the third referee report to come in, but have now decided to move ahead with the two referee reports on your study, which are included below.

As you will see, both reviewers expressed an overall interest in the study, but referee #2 does raise some major concerns that would have to be fully addressed in a revised version of the manuscript. S/he in particular points out that the contribution of Rpd3's deacetylase activity should be assessed in more detail, as well as its role as a transcriptional repressor.

Should you be able to fully address these specific concerns, as well as the various additional issues raised, then we would be happy to consider this study further for publication. I would therefore like to invite you to prepare and submit a revised manuscript.

REFeree REPORTS.

Referee #1:

This study identifies a link between RPD3 and Dot1 using very sophisticated screening strategy in yeast that had previously been developed by the same group. In depth description of genetic mutants confirms convincingly this prediction that the Rpd3L complex negatively regulated H3K79 methylation. Importantly the authors show that the same dependency exists in the mouse and further test if this also plays a function in thymic lymphomas. This reveals that reducing the dose of Dot1 increases lymphoma in an HDAC dependent manner and that Hdac1 deficient thymic lymphomas

lines require Dot1.

This is a very clearly written manuscript. Experiment and data appear to be of very high quality.

The study provides a novel chromatin link to regulation of H3K79 methylation that is conserved from yeast to man. One of its strengths is the combination of clever functional screening in a model organism and readily translating this into the complexity of cancer, where evidence for chromatin modifiers is ample but little knowledge exists how these influence tumor formation. This creates important new opportunities to gain insights into this problem and a path to use sensitive and highly controlled experiments in a model organism can generate hypothesis that can be tested in a disease setting.

Referee #2:

The manuscript by Vlaming, McLean et al. uncovers a functional relation between the histone deacetylase Rpd3 and methylation of histone H3 K79 by Dot1. They show that in *S.cerevisiae* a deletion of members of the Rpd3L complex results in increased H3K79me3 as well as H2B ubiquitination and transcription. But while H2B ubiquitination directly correlates with transcription, increased K79me3 better correlates with binding of Rpd3. They then go on to show that conditional deletion of HDAC1 in mouse thymocytes causes thymic lymphomas that depend on Dot1 dosage, as its presence prevents apoptosis.

Major concerns:

- The correlation of increased H3K79me3 in the absence of rpd3 and Rpd3 binding data makes a very compelling point for a direct regulation of H3K79 methylation by Rpd3. What is not clear is to what extent its catalytic activity as an HDAC plays a role. Given the many examples of cross-talk between histone modifications, it would be important to understand if Rpd3's regulation of H3K79 methylation is mediated by histone acetylation. (The authors refer to the cross talk between H4K16 acetylation and H3K79 methylation in their discussion, but H4K16 is not a known substrate for Rpd3 in yeast.) It therefore would be of great interest to determine if H3K79me3 is equally increased in a catalytically dead rpd3 mutant.
- Deposition of H3K79me3 has been linked to active transcription. Given the function of Rpd3 as a transcriptional repressor the role of transcription in increased K79 methylation at the 5' end of genes should include pervasive transcripts. In particular antisense transcription, as there is a direct anti-correlation between the presence of such transcripts and H3K79me3 (Brown T, et al., 2018). Do these gene groups overlap with the Rpd3-targeted gene-group in question? Do antisense transcripts change when Rpd3 is deleted? And if they do, does this depend on Dot1?
- Another anti-correlation between H3K79me3 and the INO80 complex has been reported in literature (Xue Y. et al., 2015). Again, the relation between the set of genes regulated by Rpd3 versus the one regulated by INO80 should be analyzed and discussed.

Minor concerns:

Figure 1 D and F: the y-axis title should be the same

Figure 1 E: the units should be indicated (one shouldn't need to have to go to the Methods section for this information)

Figure 2 A: the lack of increased H2Bub and H4ac of group III genes (or at least part of them) could be a technical problem due to differential strength of antibodies. This possibility should be discussed in the text; Also, the units for the y-axis should be indicated in the figure.

Figure 4 D and E: "error bars" should be called simply "bars" as they are indicating the range of 2 repeats and therefore do not show the experimental error.

Supplemental Figure 1 E: same as for Figure 1 E.

Supplemental Figure 2 D: units of the y-axis should be indicated. Also, it would be more appropriate to describe the ranking based on "gene expression" rather than "transcription", given that this does

not account for all forms of transcription.

Text:

Page 7 line 159: "To test whether H3K79me-regulated genes were direct Rpd3 targets..." This is misleading, as it could be interpreted that the expression of these genes is regulated by H3K79 methylation.

Additional suggestions:

Finally, the authors show that Hdac1-deleted thymic lymphomas are dependent on the dosage of Dot1. This raises the question whether increased H3K79 methylation is the major cause of these lymphomas or if other deacetylation driven events are necessary for tumor formation. It therefore may be interesting to determine if Dot1 overexpression mimics the effect of the deletion of HDAC1 on tumorigenesis.

1st Revision - authors' response

23rd April 2019

Referee #1:

- no comments

Referee #2:

Major concerns:

The correlation of increased H3K79me3 in the absence of rpd3 and Rpd3 binding data makes a very compelling point for a direct regulation of H3K79 methylation by Rpd3. What is not clear is to what extent its catalytic activity as an HDAC plays a role. Given the many examples of cross-talk between histone modifications, it would be important to understand if Rpd3's regulation of H3K79 methylation is mediated by histone acetylation. (The authors refer to the cross talk between H4K16 acetylation and H3K79 methylation in their discussion, but H4K16 is not a known substrate for Rpd3 in yeast.) It therefore would be of great interest to determine if H3K79me3 is equally increased in a catalytically dead rpd3 mutant.

> We thank the reviewer for this valuable suggestion. To address this question, we generated rescue constructs for rpd3 knock-out strains using wild-type RPD3 and previously described RPD3 point mutants that lack catalytic activity. We found that while re-expression of RPD3 restores the low H3K79me3 levels, the RPD3 mutants do not rescue the loss of RPD3. Therefore, we conclude that Rpd3 controls H3K79 methylation via its deacetylase activity. We describe these new results on page 7 and Fig. 2B.

Deposition of H3K79me3 has been linked to active transcription. Given the function of Rpd3 as a transcriptional repressor the role of transcription in increased K79 methylation at the 5' end of genes should include pervasive transcripts. In particular antisense transcription, as there is a direct anti-correlation between the presence of such transcripts and H3K79me3 (Brown T, et al., 2018). Do these gene groups overlap with the Rpd3-targeted gene-group in question? Do antisense transcripts change when Rpd3 is deleted? And if they do, does this depend on Dot1?

> To address whether genes with antisense transcripts overlap with the genes that show increased H3K79 methylation upon loss of Rpd3, we obtained the source data on antisense transcription from Brown et al (kindly provided by the Mellor lab) and compared it to our data. This analysis showed that Rpd3 does not specifically affect H3K79 methylation at genes with antisense transcription. Given this outcome, we did not further investigate the relation between antisense transcription and regulation of H3K79 methylation by Rpd3, also because analyzing this in more detail would require new NET-seq or PRO-seq experiments in wild-type and rpd3 mutant strains. We discuss the new analysis on page 8 and Fig. EV2E.

Another anti-correlation between H3K79me3 and the INO80 complex has been reported in literature (Xue Y. et al., 2015). Again, the relation between the set of genes regulated by Rpd3 versus the one regulated by INO80 should be analyzed and discussed.

> INO80 is indeed a very interesting regulator of H3K79me3. We compared the changes in H3K79 methylation in mutants lacking INO80 with changes in H3K79 methylation in mutants lacking

Rpd3. This analysis showed that INO80 does not specifically affect H3K79 methylation at genes that are also regulated by Rpd3. This finding is in agreement with the very large number of genes that is affected by INO80, whereas Rpd3 regulates H3K79 methylation at a subset of genes. We describe these analyses on pages 8-9 and Fig. EV2F.

Minor concerns:

Figure 1 D and F: the y-axis title should be the same

> The axes of these panels are indeed not the same; panel D is ranked on K79 me3/me1 in wild type while panel F is ranked on K79me3 in rpd3d/WT. We now emphasize in the text on page 7 that panel F represents a heatmap of H3K79me3 changes to avoid confusion.

Figure 1 E: the units should be indicated (one shouldn't need to have to go to the Methods section for this information)

> fixed; we added bars and a label to explain the units and scale.

Figure 2 A: the lack of increased H2Bub and H4ac of group III genes (or at least part of them) could be a technical problem due to differential strength of antibodies. This possibility should be discussed in the text; Also, the units for the y-axis should be indicated in the figure.

> fixed and we added a discussion on page 9.

Figure 4 D and E: "error bars" should be called simply "bars" as they are indicating the range of 2 repeats and therefore do not show the experimental error.

> fixed

Supplemental Figure 1 E: same as for Figure 1 E.

> fixed

Supplemental Figure 2 D: units of the y-axis should be indicated. Also, it would be more appropriate to describe the ranking based on "gene expression" rather than "transcription", given that this does not account for all forms of transcription.

> fixed

Text:

Page 7 line 159: "To test whether H3K79me-regulated genes were direct Rpd3 targets...." This is misleading, as it could be interpreted that the expression of these genes is regulated by H3K79 methylation.

> To avoid confusion, we rewrote this sentence as follows (line 160): 'Next, we compared the genes with H3K79me changes with published data on Rpd3 binding and H4 acetylation.'

Additional suggestions:

Finally, the authors show that Hdac1-deleted thymic lymphomas are dependent on the dosage of Dot1. This raises the question whether increased H3K79 methylation is the major cause of these lymphomas or if other deacetylation driven events are necessary for tumor formation. It therefore may be interesting to determine if Dot1 overexpression mimics the effect of the deletion of HDAC1 on tumorigenesis.

> We thank the reviewer for this suggestion. We have made some attempts to overexpress DOT1L in mammalian cells and experienced problems in obtaining stable lines. Therefore, this approach will require more optimization. One major risk of this approach is that global overexpression of DOT1L may not mimic the increase in H3K79 methylation at specific target genes in HDAC1 mutants. Given these challenges and uncertainties as well as the substantial time investment, we believe that this interesting experiment falls beyond the scope of the current manuscript.

Thank you for submitting your revised manuscript for our consideration. It has now been seen once more by the original referees, and I am pleased to say that they find that their comments have been sufficiently addressed and now support publication. However, prior to final acceptance I would ask you to address several editorial issues that are listed in detail below.

REFeree REPORTS

Referee #2:

The authors addressed all my concerns to satisfaction.

Corresponding Author Name: Fred van Leeuwen

Manuscript Number: EMBOJ-2019-101564